# Association between Maternal and Toddler Appetitive Traits in a Mexican Population

**DOI:** 10.3390/bs13100849

**Published:** 2023-10-17

**Authors:** Jocelyn González-Toribio, Claudia Hunot-Alexander, Edgar Manuel Vásquez-Garibay, Alfredo Larrosa-Haro, Erika Casillas-Toral, Carmen Patricia Curiel-Curiel

**Affiliations:** 1Instituto de Nutrición Humana, Centro Universitario de Ciencias de la Salud, Universidad de Guadalajara, Salvador Quevedo y Zubieta No. 750, Edificio Anexo al Hospital Civil “Dr. Juan I. Menchaca”, Piso 3, Guadalajara C.P. 44340, Jalisco, Mexico; jocelyn.gonzalez7445@alumnos.udg.mx (J.G.-T.); edgar.vgaribay@academicos.udg.mx (E.M.V.-G.); alfredo.larrosa@academicos.udg.mx (A.L.-H.); carmen.curiel@academicos.udg.mx (C.P.C.-C.); 2Hospital Civil Juan I. Menchaca, Salvador Quevedo y Zubieta 750, Guadalajara C.P. 44340, Jalisco, Mexico; erika.casillas@academicos.udg.mx; 3Licenciatura en Nutrición, Centro Universitario del Sur, Universidad de Guadalajara, Av. Enrique Arreola Silva No. 883, Colonia Centro, Ciudad Guzmán C.P. 49000, Jalisco, Mexico

**Keywords:** toddlers, appetite, appetitive traits, BMIz score

## Abstract

The Child Eating Behaviour Questionnaire for toddlers (CEBQ-T-Mex) and the Adult Eating Behaviour Questionnaire (AEBQ-Esp) measure appetitive traits (ATs) in children and adults, respectively, both validated for use in Spanish. ATs are inherited variations in appetite, present from birth, that are reasonably stable throughout childhood and can explain why some infants over- or undereat in response to environmental exposures. “Food approach” traits predispose to overweight while “food avoidance” traits provide protection, but little is known about the relationships between parents’ and their toddler’s ATs. The aim was to examine the associations between maternal and toddler appetitive traits, using the AEBQ-Esp and CEBQ-T-Mex, and to examine the associations between ATs and Body Mass Index z-scores (BMIz). Sociodemographic data and the weights and heights of mothers and toddlers (aged 12–36 months) were collected from a teaching hospital in Guadalajara, Mexico. Mothers completed both the AEBQ-Esp and the CEBQ-T-Mex. Direct correlations were found between the ATs of toddlers and their mother (*p* < 0.005), except for “Slowness in Eating” (SE), and only an inverse correlation was found between the “Satiety Responsiveness” (SR) of toddlers and their BMIz (r = −0.147; *p* = 0.007). These results suggest that ATs could potentially run in families. These may be useful targets for family-wide interventions to support the development and maintenance of healthy eating behaviours in childhood.

## 1. Introduction

In the past two decades, the global prevalence of obesity has surged significantly, emerging as a prominent concern within public health. Notably, toddlers represent a stage in the life course that can be particularly susceptible, marked by their initiation into family mealtimes. This transitional phase introduces shifts in eating behaviours and psychobehavioural aspects of the feeding environment that become important [1].

The Behavioural Susceptibility Theory (BST) proposes that susceptibility to obesity has a genetic basis, and results from an interaction between an individual’s genetic risk and their environmental exposure [2,3]. Central to the BST is the hypothesis that inherited differences in appetite act as behavioural mediators of an individual’s genetic susceptibility to the “obesogenic” environment. Individual differences in appetite determine why some people overeat and others do not in response to environmental opportunities [2]. Two neurologically dissociable aspects of appetite are thought to be involved: responsiveness to food cues (e.g., wanting to eat in response to the sight, smell, or taste of food), which reflects hedonic processes involved in pleasure and reward; and sensitivity to internal satiety signals (e.g., feelings of fullness), which expose homeostatic processes involved in energy balance [4]. In contrast, those who are genetically predisposed to high satiety sensitivity or low interest in food are protected from obesity, or even at risk of underweight, under the same environmental pressures to eat [2]. For example, individuals who are predisposed to be more responsive to food cues and less sensitive to satiety signals are more likely to overeat in response to the modern obesogenic food environment that presents increased access to the cheap, highly palatable, energy-dense foods promoted by the food industry in countries such as Mexico [5].

To provide evidence for the BST, several comprehensive psychometric reliable measures of appetite were developed to assess appetitive traits, described as behavioural tendencies towards food and eating occasions. These measures evaluate similar appetitive traits, including “Food Responsiveness” and “Satiety Responsiveness”. They have been adapted for use in different stages of the life course, such as the Baby Eating Behaviour Questionnaire in 0- to 6-month-old milk-fed infants [6,7], the Child Eating Behaviour Questionnaire for use in Toddlers (CEBQ-T) aged 1 to 3 years [7,8], and the CEBQ in children aged 3 to 13 years [9,10]. Appetitive traits have also been measured in adults using the Adult Eating Behaviour Questionnaire (AEBQ) [11], previously validated for use in a Mexican sample (Adult Eating Behaviour Questionnaire—Español: AEBQ-Esp) [12], which can be used to assess maternal traits.

Evidence to support the BST has been provided in a systematic review of articles using the BEBQ (n = 5) and CEBQ (n = 67), which provides a preliminary analysis that supports the hypothesis that the greater the weight, the greater the presence of food approach traits (e.g., “Food Responsiveness”) and the lower the weight, the greater the presence food avoidance traits (e.g., “Satiety Responsiveness”) [13]. Also, twin studies have been used to show the relative genetic contributions of appetite to variations in weight in infants [14,15] and into childhood [16]. These have shown that differences in weight may be partly due to genetically determined differences in appetitive traits that confer differential susceptibility to obesogenic environments [4].

However, although there is sufficient evidence of the genetic link between appetite and weight variations, this is not likely the only pathway to express changes in weight, given that the correlations observed are small [4]. Many other factors could be contributing to weight trajectories in infants, toddlers, and children and, further, play a role in the intergenerational transmission of appetite and eating behaviours [17]. Genetic and environmental preferences for different foods have also been observed and could contribute to different weight trajectories in infants and children, with genetic effects playing a bigger role in the preference for sweeter-tasting fruits, bitter-tasting vegetables, and proteins in a large sample of 3-year-old British twins (n = 2686) [18].

Also, the parent’s affective and personality disorders could undermine the functioning of the caregiving relationship, resulting in unpredictable and inconsistent emotional communication with the child [17]. Parents are one of the most dominant influences on their children’s eating, particularly in early life [19]. Parental feeding practices are a key component of a child’s food environment. Evidence suggests that the use of nonresponsive feeding practices have a detrimental impact on appetite development, by encouraging children to eat for reasons other than hunger (i.e., using food as a reward for desired behaviour or to soothe emotions) or may override their satiety cues by pressuring them to eat [19,20]. Also, they may influence them in multiple ways, including via modelling of eating behaviours; however, the relationship between parent and child self-regulation of appetitive traits has received little attention [21]. Responsive feeding practices by mothers were better achieved when toddlers had higher “Satiety Responsiveness” and lower “Food Responsiveness” [22]. When children have avid appetites, parents may also find it difficult to maintain a healthy food environment as their demands for certain foods to be present in the home can be stressful and occur as early as the introduction of family meals during early toddlerhood [19].

Parents may also find it difficult to recognise their own styles of eating and disordered eating behaviours that may have an impact on their child’s eating behaviours or their ability to diagnose their own child’s style of eating, mediating parent–infant interactions [17,23]. Other psychophysical aspects of eating behaviour could also be playing a part in the influence of appetite on weight. For example, both pre- and post-gestational differences in maternal metabolism modify the mother’s eating behaviour, which, in turn, could affect the infant’s appetite as well as taste and odour for different types of foods [24].

Appetitive traits and parental practices have been studied in children aged 2 to 5 years in Australia, with maternal feeding practices and child “Satiety Responsiveness” and “Food Responsiveness” showing strong continuity over time, from 2 to 5 years old; maternal feeding practices were also found to be associated with child “Food Responsiveness” over time. Conversely, child “Satiety Responsiveness”, but not “Food Responsiveness”, was also associated with maternal feeding practices over time. These results are consistent with interventions that provide feeding advice to parents on how to respond appropriately to individual child eating behaviour phenotypes [25]. Infant weight differences may be partly due to genetically determined appetitive traits that confer susceptibility to obesogenic environments [15].

Thus, many factors play a potential a role in the predisposition of toddlers to develop overweight or obesity. However, the results from a cross-sectional study conducted in a Mexican sample of mother–child dyads (aged 5–12) showed the relationships between maternal and child appetitive traits. Small to moderate positive associations between maternal and child appetitive traits (“Food Responsiveness” (r = 0.22; *p* < 0.001); “Emotional Overeating” (r = 0.30; *p* < 0.001); “Enjoyment of Food” (r = 0.15; *p* < 0.001); “Satiety Responsiveness” (r = 0.16; *p* < 0.001); “Emotional Undereating” (r = 0.34; *p* < 0.001), and “Food Fussiness” (r = 0.14; *p* < 0.001)) were observed. No association was observed between maternal and child “Slowness in Eating” (r = 0.01; *p* > 0.05) [10]. An intergenerational relationship could exist between the mother’s and toddler’s appetitive traits; however, further research is required in this area.

The present study has several aims: (1) to use the AEBQ-Esp (Mexican Spanish validated version) [12] and the CEBQ-T validated for a Mexican population (CEBQ-T-Mex) [7] to examine the relationships between maternal and child appetitive traits. We hypothesise that there is a positive association between mothers and their children’s corresponding appetitive traits for food approach and food avoidance appetitive traits and (2) to examine the associations between appetitive traits and Body Mass Index z-scores (BMIz) in toddlers.

## 2. Materials and Methods

### 2.1. Participants and Procedures

The participants were recruited from an outpatient clinic at the Pediatric Department of the Nuevo Hospital Civil de Guadalajara Juan I Menchaca (NHCJIM) from May 2019 to March 2020. Three hundred and thirty mother–toddler dyads who met the inclusion criteria and agreed to sign the informed consent completed the baseline questionnaires. The participants were recruited through consecutive case sampling. The main exclusion criteria were: (i) toddlers with diseases that could modify their appetites, such as acute respiratory or digestive diseases of more than a week of evolution, malnutrition, major congenital malformations, or chronic diseases; (ii) toddlers who did not attend the hospital with their mothers; and (iii) mothers who were pregnant and/or breastfeeding, or with a diagnosed personality or eating disorder. Toddlers from 12 to 36 months of age, who were born with a healthy gestational age, weight, and height and who did not present pathological complications that interfered with their eating behaviours, in addition to being accompanied by the biological mother were included in the study. Mothers’ inclusion criteria included women aged 18 to 40 years old, conscious of their children’s characteristics. A team of nutrition graduates from the University of Guadalajara collected data over a 6-month period at the NHCJIM outpatient clinic. They explained the study’s objectives to the patients, obtained biological kinship and informed consent, and measured the mothers’ anthropometric data. Then, they asked the mothers to complete the AEBQ-Esp and measured the anthropometric data for toddlers. Finally, they were asked to complete the CEBQ-T. In appreciation, the team provided information to improve toddler dietary habits and tools for healthy feeding practices.

### 2.2. Sociodemographic Variables

Sociodemographic data were collected and included maternal level of education (basic level (minimum of six years of schooling or less); secondary education (minimum of a baccalaureate or technical level of education); professional education (bachelor’s education or more)); mother’s occupation (housewife or employed); marital status (in partnership (married or cohabiting); single (single, divorced, separated, or widowed)). In addition, family type was examined and classified as “nuclear” when only parents and children lived under the same roof and as “other” when there was someone else living at home. Mothers also reported their monthly income, which was dichotomised as less than or equal MXN 4500 (equivalent to USD 264, approximately MXN 0.17/USD 1 in August 2023) or more than MXN 4500. Toddlers’ age and sex were also reported by the mothers.

### 2.3. Anthropometry

Anthropometric data were collected. The measurement of weight and height in toddlers was chosen because they are simple, safe, and non-invasive procedures, which allow for precise and accurate data using standardised protocols and appropriate equipment. Toddlers were weighed with a ^®^Tanita scale [26], and height over two years of age was measured using a stadiometer ^®^SECA213 [27]. Length and height were measured in centimetres and weight in grams. These measurements were used to calculate BMIz and classify them according to WHO standards (normal/healthy weight: BMIz = 0.9–1.9; overweight: BMIz = 2.0–2.9; obesity: BMIz ≥ 3) [28].

The weights and heights of the mothers were measured using standardised instruments (Tanita scale and SECA stadiometer) [26,27]. Height was recorded in centimetres and weight in kilograms (to the nearest 100 g), and subsequently used to calculate Body Mass Index (BMI) (kg/m^2^). Anthropometric data were used to categorise mothers’ weight statuses (healthy weight, overweight, and obese) [29].

### 2.4. AEBQ and CEBQ-T-Mex Questionnaires

The mothers’ appetitive traits were measured using the validated Spanish translation of the AEBQ, referred to as the AEBQ-Esp [12]. The AEBQ-Esp is a self-report questionnaire of 30 items that captures seven appetitive trait subscales and three food approach subscales: “Food Responsiveness” (four items: e.g., “I am always thinking about food”), “Emotional Overeating” (five items: e.g., “I eat more when I’m upset”), and “Enjoyment of Food” (three items: e.g., “I love food”). It also measures the same four “food avoidance” subscales: “Satiety Responsiveness” (four items: e.g., “I get full up easily”), “Emotional Undereating” (five items: e.g., “I eat less when I´m worried”], “Food Fussiness” (five items: e.g., “I refuse new foods at first”), and “Slowness in Eating” (four items: i.e., “I eat slowly”). Items are answered on a five-point Likert scale (1 = strongly disagree, 2 = disagree, 3 = neither agree nor disagree, 4 = agree, 5 = strongly agree). The internal reliability was good in this validation for all subscales (Cronbach’s α = 0.70–0.86).

The mothers reported their toddlers’ appetites using the 26-item CEBQ-T-Mex [4]. The CEBQ-T-Mex measures three food approach subscales (“Food Responsiveness”, “Enjoyment of Food”, and “Emotional Overeating”) and three food avoidance subscales (“Satiety Responsiveness”, “Food Fussiness”, and “Slowness in Eating”). Both questionnaires have a Likert response scale of 1 to 5 (1 = Never, 2 = Rarely, 3 = Sometimes, 4 = Often, 5 = Always). The internal reliability was good in this validation for all subscales (Cronbach’s α = 0.64–0.84).

### 2.5. Statistical Analysis

#### 2.5.1. Descriptive Statistics and Missing Data

All statistical analyses were completed using SPSS version 25. The means and standard deviations of each subscale for both the CEBQ-T and the AEBQ were obtained by calculating the average score of the items corresponding to each subscale. Each item’s skewness and kurtosis values were examined for the AEBQ and CEBQ-T-Mex subscales and were found not to be normal. However, due to the sample size and based on the central theorem, parametric tests were used.

#### 2.5.2. Associations between Sociodemographic Characteristics and Toddlers’ and Mothers’ Appetitive Traits

A T-test and one-way ANOVA were used to find associations between toddlers’ appetitive traits (“Food Responsiveness”, “Emotional Overeating”, “Enjoyment of Food”, “Satiety Responsiveness”, “Food Fussiness”, and “Slowness in Eating”) and toddlers’ sex (female and male), age category (12–24 and 24–36 months), and feeding type (breastfeeding, mix, and formula). Moreover, we calculated the Pearson correlations between the toddlers’ appetitive traits and sociodemographic characteristics (age, BMIz), and to assess the relationships between the toddlers’ and mothers’ appetitive traits (“Food Responsiveness”, “Emotional Overeating”, “Enjoyment of Food”, “Satiety Responsiveness”, “Food Fussiness”, and “Slowness in Eating”).

## 3. Results

### 3.1. Descriptive

The sociodemographic characteristics of 330 mother–toddler dyads are presented in Table 1. The majority of the mothers were aged between 18 and 25 years (48.0%), had a basic level of education (60.3%), were housewives or unemployed (75.8%), had a monthly family income below MXN 4500, equivalent to less than USD 264 (82.6%), and were raising their children in a nuclear family (63%). The children were, on average, 21 months old, 55.8% were male, most had a healthy BMIz range (94%), and 34.5% were second children.

### 3.2. Appetitive Trait Means by Sex, Age, and Feeding Mode in the First Months of Life

The means and standard deviations for each CEBQ-T subscale are shown by sex in Table 2, by age in Table 3, and by feeding mode in the first months of life in Table 4. There were no significant results according to appetitive traits by sex. Toddlers aged 24–36 months had lower “Enjoyment of Food” (*p* = 0.013) and higher “Food Fussiness” (*p* = 0.004) than 12–24-month-olds. Differences in breast-fed toddlers were found: they had less “Slowness in Eating” than mix-fed (*p* = 0.021) and bottle-fed toddlers (*p* = 0.015).

### 3.3. Associations between Appetite Traits, Age and BMIz Scores

The bivariate correlations between the appetite traits, age, and BMIz of the toddlers, by age categories of the mother, are shown in Table 5. We found that “Food responsiveness” and “Enjoyment of Food” were inversely correlated with age (*p* = 0.037 and *p* = 0.001) and directly with “Food Fussiness” (*p* = 0.000). Regarding the toddlers’ BMIz, we only found an inverse correlation with “Satiety Responsiveness” (*p* = 0.007). These associations were not seen when maternal appetitive traits were separated by age categories.

### 3.4. Associations between Toddler´s and Mother’s Appetitive Traits

To show the relationships between the mothers’ and toddlers’ appetitive traits, the correlations between concordant subscales were examined. As hypothesised, the maternal appetitive traits were significantly and positively correlated with their child´s equivalent appetitive traits, except for “Slowness in Eating” (r = 0.01; *p* > 0.05) (Table 5). Correlations were of small to moderate magnitude (“Food Responsiveness” (r = 0.20; *p* < 0.001), “Emotional Overeating” (r = 0.19; *p* < 0.001), “Enjoyment of Food” (r = 0.20 *p* < 0.0001), “Satiety Responsiveness” (r = 0.12; *p* < 0.05) and “Food Fussiness” (r = 0.16; *p* < 0.001)). These associations were similarly observed by maternal age category in the 18–25-year-olds (“Food Responsiveness” (r = 0.25; *p* < 0.001), “Emotional Overeating” (r = 0.28; *p* < 0.001), “Enjoyment of Food” (r = 0.22 *p* < 0.0001), “Satiety Responsiveness” (r = 0.22; *p* < 0.001) and “Food Fussiness” (r = 0.25; *p* < 0.001)), as well as for “Food Responsiveness” (r = 0.25; *p* < 0.001), “Emotional Overeating” (r = 0.19; *p* < 0.001) and “Enjoyment of Food” (r = 0.27; *p* > 0.001) in the 31–40-year-olds.

## 4. Discussion

This is the first study to be conducted in a low socioeconomic Mexican sample of mother–toddler dyads that explores the relationships between maternal and child appetitive traits, indicating that all appetitive traits measured in mothers were positively correlated with their child’s corresponding appetitive traits, except for “Slowness in Eating”, when all mothers were grouped into one category.

During the first years of life, children and caregivers learn to recognise and interpret both verbal and nonverbal communication signals from one another [30]. This reciprocal process forms a basis for the emotional bonding or attachment between children and caregivers that is essential to healthy social–emotional functioning [31]. The presence of affective and personality disorders in parents may disrupt the functioning of the caregiving relationship, leading to erratic and inconsistent emotional communication with their child [17], affecting developmental weight trajectories in children observed longitudinally [32]. However, no mothers with a known history of personality or eating disorders were included in the study.

This study is the first to evaluate the appetitive traits measured psychometrically, used to strengthen the BST, in the mother–toddler dyad in a low sociodemographic Mexican population. As hypothesised, most maternal appetitive traits were positively and significantly associated with the corresponding subscales of their toddler counterpart, except for “Slowness in Eating”. These same associations were not found when the mother’s age was categorised, except in those mothers who were the youngest (18–25 years of age) and, also, the highest in number in our study and only some associations were found in the 31- to 40-year-old group, where only “Food Responsiveness”, “Emotional Overeating”, and “Enjoyment of Food” were positively correlated with their toddler’s counterpart appetitive traits. These differences could be due, in part, to the metabolic and neurological conditions associated with appetite in different age groups of the mother [24].

To date, there have been no studies in adults that have analysed the associations between appetitive traits and BMI by different age groups. From birth, appetite differences have been linked to weight gain rates and previous research has explored the developmental progression of appetitive traits [14,15,16]. Recently, a study tracking children from ages 4 to 11 showed that modest to moderate positive correlations emerged in their appetitive traits over time. These correlations involved an increase in food-approach traits and a decrease in food-avoidant traits, although the findings regarding changes in their “Enjoyment of Food” were inconsistent [16]. However, no studies have investigated these trajectories into adulthood.

In this study, “Slowness in Eating” was the only trait that was not significantly correlated between mothers of all ages and toddlers, a result that corresponds with those observed in the mother–child dyad (n = 842; aged 3–13 years) of similar socioeconomic backgrounds at the HCGJIM [10].

### 4.1. Toddlers’ Characteristics and Appetitive Traits

In previous studies, appetitive traits in infants differ by sex; male infants have higher “Food Responsiveness” and “Appetite”, and lower “Satiety Responsiveness” [7,33]. In Australia, male infants have shown a greater presence of the “Slowness in Eating” trait [34]. In this study, no significant correlations were found between sex, possibly because there is no difference in energy needs by sex until four years of age [35].

It is well known that energy demands decrease in the toddler stage [36] and characteristics such as food neophobia and picky eating begin to appear [37]. This is consistent with our findings, which highlight that “Enjoyment of Food” decreased from 12 to 24 months and “Food Fussiness” increased over this period.

In this study, we also observed that the mode of feeding in the first six months of life may also play a role, with “Slowness in Eating” found to be higher in toddlers who were exclusively breastfed compared to those who consumed breastmilk and/or formula. This points to a different benefit of long-term breastfeeding that was shown in a longitudinal study in Ireland in which 230 mother–infant dyads from two months to five years were evaluated; they reported that children who received exclusive breastfeeding for a longer time during the first months of life (more than six months) had lower “Food Responsiveness” [38].

### 4.2. Appetitive Traits and BMIz

Furthermore, we observed an inverse correlation between “Satiety Responsiveness” and BMIz. This result corresponds with the results found in a prospective study in Australia that was carried out using a sample of 37 mother–infant dyads. “Satiety Responsiveness” was shown to be inversely correlated with the BMIz of infants [34]. Also, in Chile, it was observed that younger 5-month-old infants with obesity had lower “Satiety Responsiveness” compared with overweight and healthy-weight infants [33]. The correlation between appetitive traits and BMIz have also been studied in other countries, such as Wales in 298 toddlers aged 18 to 24 months [39] and in China in 219 children aged 12 to 18 months [40]; however, they did not show significant correlations [13]. In the Mexican study, appetitive trait correlations with BMIz in school aged children showed that “Food Responsiveness”, “Emotional Overeating”, and “Food Fussiness” were significantly positively associated with BMIz scores. On the other hand, “Satiety Responsiveness”, “Emotional Undereating”, and “Slowness in Eating” were significantly and negatively associated with BMIz scores in these children [10]. This may indicate that, at the toddler stage, external obesogenic factors such as the food environment have not yet had a significant influence on the toddler’s eating behaviours and therefore it is not reflected in the toddler’s weight status [16].

We also observed that the correlation between “Satiety Responsiveness” and BMIz goes from negative to positive when adjusting for maternal appetite and “Enjoyment of Food”, and BMIz goes from positive to negative when adjusting for maternal appetite. This could be because the mother influences their toddler’s eating by modelling their own eating behaviours, which could be reflected in the BMIz [41].

### 4.3. Maternal and Toddler Appetitive Traits

As with the results observed in a population of mother–child dyads of 3- to 13-year-old children of the same socioeconomic backgrounds, collected at the same public hospital (the HCJIM), an association suggesting a potential intergenerational transmission of appetite was found between the same mother–toddler dyads, as with the mother–child dyads of slightly older children. It would appear, therefore, that “Slowness in Eating” does not appear to be as closely related between the dyads at both of these stages of the life course. For example, certain food choices, such as consumption, monitoring, conditioning of treats, reinforcing of food rules, and restricting food choices, have been seen in the qualitative study of the intergenerational transmission of eating behaviour through three generations in Anglo–Australian (n = 11), Chinese–Australian (n = 8), and Italian–Australian (n = 8) families (n = 114) [34]. This suggests that eating at a fast or slow pace may be a cultural state that is not genetically transmitted.

Parents not only create food environments for children’s early experiences with food and eating, but they also influence their children’s eating by modelling their own eating behaviours, taste preferences, and food choices [41]. Many research reviews have focused on the robust body of evidence on coercive control in feeding: how parenting practices such as restriction and pressure to eat increase children’s risk of developing undesirable eating behaviours and unhealthy weight outcomes. However, parents’ own eating behaviours also have an impact on their children’s wellbeing and emotional and behavioural issues [42] and these issues need to be included in these studies, but were not part of the aims here. 

Current evidence supports starting with responsive feeding and parenting during infancy and incorporating structure and limit-setting in early childhood, with monitoring and mealtime structure remaining important during middle childhood and adolescence [43]. Coercive feeding practices such as the use of pressure to eat or using food as a reward should be avoided, as these can create negative associations with the food or meals and lead to food refusals. Instead, caregivers can model eating and enjoying the food. Non-food rewards, such as praise or stickers, can also be used to encourage children to taste a food without negative outcomes [44]. Appetitive traits should be used to tailor interventions in these age groups [45].

This is a novel study that measures appetite traits in mother–toddler dyads, which suggest there is the potential for an intergenerational transmission of appetite. These associations and the measurement of many other factors at play, not present in this study, could provide some evidence towards observed inherited variations in appetite [4] between mothers and toddlers. Thus, the study presents several limitations. The mothers and their toddlers who took part in the study were recruited from a large public hospital and came from homogeneous sociodemographic families, with very similar education levels, and average family monthly incomes [7,10]. Therefore, there is a need to replicate this study in more diverse populations, with the potential to observe different correlations between the toddlers’ appetitive traits and BMIz in the participants and possibly to see changes in BMIz in the toddlers. Therefore, it is not possible to generalise the results, given the limited sociocultural background of the mothers, whose answers to the questionnaires might have resulted from a problematic understanding of the questionnaire items [7]. Also, a lack of the measure of the psychopathology of the mother prevents the development of an understanding of whether a maternal background of disordered eating behaviour could be influencing the toddler’s own eating behaviour [17,23]. The measurement of appetitive traits in the mother before the child was born would also be necessary. Therefore, future longitudinal research that explores the bidirectional relationship between appetitive traits and BMI from infancy to adulthood may help confirm whether appetitive traits play a causal role in weight gain, particularly during early development [16]. These studies should explore the relationship between maternal and child appetitive traits and study their mediating effect on the risk of overweight and obesity. Further studies are needed to develop interventions that help mothers of toddlers to better incorporate the recognition of hunger and satiety cues and develop interventions that help food choices, modelling of eating behaviour, and changes in body composition and nutritional status [46], as well as a measure of a number of different complex variables involved in psychopathological risk through intergenerational transmission [17], or as risk factors for disordered eating development through the life course [47].

## 5. Conclusions

This appears to be the first study to measure the relationships between maternal and toddler appetitive traits, conducted in a low socioeconomic Mexican population, showing that the appetitive traits measured in mothers are positively correlated with their toddler’s corresponding appetitive traits. This was true for all food approach appetitive traits, such as “Food Responsiveness”, “Emotional Overeating”, and “Enjoyment of Food”, and two out of three food avoidance appetitive traits, “Satiety Responsiveness” and “Food Fussiness”, but not “Slowness in Eating”, when all maternal ages were taken into account, but not when they were categorised by age. These results point towards a potential intergenerational transmission of appetite, which, as explained by the BST, acts as a mediator for the genetic influence on weight through the presence of certain appetitive traits with a strong genetic basis. However, these are not the only factors that are at play, given the small to moderate magnitude of the correlations presented.

Furthermore, “Food Responsiveness” and “Enjoyment of Food” were inversely correlated with the toddlers’ age and were directly correlated with “Food Fussiness”. Toddlers’ BMIz were found to be inversely correlated with “Satiety Responsiveness”, yet no other correlation between the toddlers’ appetitive traits and BMIz were found. These results point towards the changes that occur through time that are part of the gene–environment interaction. Once the toddler becomes older, certain appetitive traits are further established and further correlations with BMIz can be seen, as is the case with older children observed in other studies.

The results from this study support the use of family interventions that target appetitive trait profiles that help improve eating and health outcomes in toddlers with differing appetites. The recognition of the mother’s appetitive traits, as well their toddler’s, could help install responsive feeding practices that encourage modelling strategies and the self-regulation of eating behaviour.

## Figures and Tables

**Table 1 behavsci-13-00849-t001:** Characteristics of mother–toddler dyads (n = 330).

Characteristics	M (SD) or n (%)
Maternal	
Maternal age (years)	26.8 (6.1)
18–25 years	153 (48.0)
26–30 years	79 (24.8)
31–40 years	87 (27.3)
Relationship status	
Married/cohabitation	266 (80.6)
Divorced/Separated/Single	64 (19.4)
Education	
Basic education	199 (60.3)
Secondary education	92 (27.9)
Professional education	39 (11.8)
Employment	
Housewife/Unemployed	250 (75.8)
Employed	80 (24.2)
BMI (Kg/m^2^)	26.5 (5.5)
Child	
Gender	
Male	184 (55.8)
Female	146 (44.2)
Age months	21 (6.5)
12–24 months	230 (69.7)
24–36 months	100 (30.3)
z-BMI	0.2 (1.1)
Healthy	307 (94.0)
Overweight	9 (2.5)
Obesity	11 (3.5)
Birth order	
Only child	107 (32.7)
First child	17 (5.2)
Second child	114 (34.9)
Other	89 (27.2)
Feeding type in the first months	
Breastfeeding	148 (45.0)
Mix	113 (34.0)
Formula	69 (21.0)
Family type	
Nuclear	208 (63.0)
Other	122 (37.0)
Monthly family income	
≤MXN 4500	265 (82.6)
>MXN 4500	56 (17.4)

**Table 2 behavsci-13-00849-t002:** Means and standard deviations for each scale of the CEBQ-T by sex.

Variables	Males		Females		*p*
	M ^1^	SD ^2^	M ^1^	SD ^2^	
Food Responsiveness	2.5	1.2	2.6	1.3	0.494
Emotional Overeating	1.3	0.6	1.5	0.8	0.064
Enjoyment of Food	3.9	1.0	3.8	1.0	0.351
Satiety Responsiveness	2.7	1.0	2.9	0.9	0.058
Food Fussiness	2.4	1.1	2.4	1.0	0.814
Slowness in Eating	3.0	0.5	2.9	0.6	0.444

^1^ mean; ^2^ standard deviation.

**Table 3 behavsci-13-00849-t003:** Means and standard deviations for each scale of the CEBQ-T by age.

Variables	12–24 Months		24–36 Months		*p*
	M ^1^	SD ^2^	M ^1^	SD ^2^	
Food Responsiveness	2.6	1.2	2.5	1.3	0.440
Emotional Overeating	1.4	0.6	1.5	0.7	0.250
Enjoyment of Food	4.0	0.9	3.7	1.0	0.013 ^3^
Satiety Responsiveness	2.7	1.0	2.8	0.9	0.501
Food Fussiness	2.3	1.0	2.7	1.2	0.004 ^4^
Slowness in Eating	3.0	0.5	3.1	0.6	0.055

^1^ mean; ^2^ standard deviation; ^3^ the media analysis shows that the “Enjoyment of Food” is higher at 12–24 months than at 24–36 months; ^4^ the media analysis shows that the “Food Fussiness” is higher at 26–36 months than at 12–24 months.

**Table 4 behavsci-13-00849-t004:** Means and standard deviations for each scale of the CEBQ-T by feeding mode in the first months of life.

Variables	Breast-Fed		Mixed-Fed		Bottled-Fed		*p*
	M	SD	M	SD	M	SD	
Food Responsiveness	2.5	1.2	2.5	1.3	2.6	1.2	0.696
Emotional Overeating	1.3	0.6	1.5	0.7	1.5	0.7	0.232
Enjoyment of Food	4.0	0.9	3.8	1.0	3.9	1.0	0.401
Satiety Responsiveness	2.8	1.0	2.7	1.0	2.9	0.9	0.572
Food Fussiness	2.3	1.1	2.5	1.1	2.6	1.1	0.167
Slowness in Eating	2.9	0.5	3.1	0.5	3.1	0.6	0.004 ^1^

^1^ The one-way ANOVA shows significant differences in SE by feeding mode; Bonferroni post hoc testing shows that breast-fed toddlers had lower SE than mixed-fed (*p* = 0.021) or bottled-fed (*p* = 0.015) toddlers.

**Table 5 behavsci-13-00849-t005:** Correlation coefficients of toddler appetitive traits with toddlers’ age, toddlers’ BMIz, and the corresponding mothers’ AEBQ subscales by age group.

Mother		Toddler							
				(CEBQ-Mex)					
Age	AEBQ-Esp	Age	BMIz	FR	EOE	EF	SR	FF	SE
**18–25 years** ** n = 153** ** (48%)**	**FR**	−0.073	0.091	0.250 **	0.181 *	0.193 *	0.172 *	0.006	−0.067
	**EOE**	0.017	0.019	0.106	0.280 **	0.055	0.020	0.102	0.123
	**EF**	0.085	0.082	0.229 **	0.092	0.217 **	−0.106	−0.180 *	0.086
	**SR**	0.080	0.009	−0.004	0.072	−0.101	0.222 **	0.176 *	0.066
	**EUE**	−0.052	0.102	0.017	0.049	0.068	0.046	0.034	0.075
	**FF**	−0.002	−0.098	−0.098	−0.065	−0.114	0.098	0.249 **	−0.047
	**SE**	−0.056	−0.024	0.212 **	0.055	0.152	0.001	−0.186 *	−0.001
**26–30 years** ** n = 79** ** (24.8%)**	**FR**	−0.028	−0.015	0.059	−0.091	−0.073	0.119	0.130	−0.018
	**EOE**	0.101	−0.010	−0.004	0.003	−0.117	0.262 *	0.089	0.063
	**EF**	−0.003	−0.026	0.044	0.090	0.150	0.077	−0.020	−0.011
	**SR**	0.025	0.084	0.117	0.051	−0.015	−0.036	0.249 *	0.109
	**EUE**	−0.158	−0.068	0.153	−0.139	0.009	−0.083	0.011	0.035
	**FF**	0.014	0.163	−0.034	−0.074	−0.158	0.029	0.166	0.032
	**SE**	0.148	0.118	0.003	−0.043	0.001	0.146	0.105	−0.113
**31–40 years** ** n = 87** ** (27.3%)**	**FR**	−0.089	0.078	0.248 *	0.189	0.172	0.072	0.138	−0.072
	**EOE**	−0.016	0.016	0.126	0.248 *	0.065	0.112	0.192	0.006
	**EF**	0.077	−0.110	0.145	0.031	0.267 *	−0.063	0.130	−0.007
	**SR**	0.123	0.047	−0.014	0.051	−0.183	0.161	0.191	0.016
	**EUE**	−0.134	−0.116	0.077	0.113	−0.004	0.012	−0.171	0.124
	**FF**	−0.037	−0.103	0.141	0.181	0.059	−0.045	0.049	0.167
	SE	0.079	0.029	0.118	0.034	0.070	−0.003	−0.009	0.124
**18–40 years** ** n = 319** ** (100%)**	**FR**	−0.072	0.060	0.208 **	0.129 *	0.123 *	−0.023	0.075	−0.068
	**EOE**	0.162 **	0.017	0.062	0.192 **	0.009	0.124 *	0.127 *	0.066
	**EF**	−0.044	−0.005	0.157 **	0.078	0.205 **	−0.042	−0.071	0.026
	**SR**	−0.108	−0.031	0.041	0.065	−0.080	0.126 *	0.187 **	0.077
	**EUE**	−0.034	0.005	0.059	0.025	0.054	0.017	0.039	0.017
	**FF**	−0.143 **	−0.120 *	0.000	0.001	−0.078	0.039	0.165 **	0.014
	**SE**	−0.061	0.010	0.153 **	0.050	0.105 *	0.017	−0.084	0.010

FR = Food Responsiveness; EOE = Emotional Overeating; EF = Enjoyment of Food; SR = Satiety Responsiveness; EUE = Emotional Undereating; FF = Food Fussiness; SE = Slowness in Eating; AEBQ-Esp = Adult Eating Behaviour Questionnaire, Spanish version; CEBQ-Mex = Child Eating Behaviour Questionnaire, Mexican Spanish version. * Correlation is significant at the 0.05 level (two-tailed); ** Correlation is significant at the 0.001 level (two-tailed).

## Data Availability

The datasets used and/or analysed during the current study are available from the corresponding author upon reasonable request.

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
