# Peer review of "Association between Maternal and Toddler Appetitive Traits in a Mexican Population"

_behavsci, 2023, doi:10.3390/bs13100849_

Round 1

Reviewer 1 Report

This is an interesting paper addressing an important topic and potentially adding to previous literature. It also fits the journal's scope.

I have some suggestions to improve the manuscript before it can be considered for publication, as follows:

I think a separate section "The present study" would help the reader understand and locate better aims and hypotheses.

It is not clear to me which the inclusion criteria for children were. Please, clarify.

Who recruited the sample? How? Were the mothers compensated for their recruitment?

Why did the authors choose these measures over others?

In the results section the alert "Error! Reference source not found" is present several times.

My main concern, which applies both to the introduction and discussion section is that there is very little or none reference to the Developmental Psychopathology theoretical and clinical framework. In general, the manuscript lacks a developmentally oriented standpoint. Besides, the aspects associated with genetic issues are also scarcely addressed. I suggest adding information coming from the following references: https://www.cairn.info/revue-devenir-2004-3-page-173.htm ; https://doi.org/10.1002/imhj.21732 ; https://www.ncbi.nlm.nih.gov/pmc/articles/PMC4726243/

The quality of English is good. I only saw minor typos and/or grammatical errors.

Author Response

This is an interesting paper addressing an important topic and potentially adding to previous literature. It also fits the journal's scope.

I have some suggestions to improve the manuscript before it can be considered for publication, as follows:

As per your suggestions and thorough and constructive review, we have made suggested changes and highlighted them in the document itself. We address each of your points below.

I think a separate section "The present study" would help the reader understand and locate better aims and hypotheses.

Thank you for your insightful suggestion. We have created a separate paragraph, where we have included the aims and hypothesis of the study, in line 132: “The present study……”.

It is not clear to me which the inclusion criteria for children were. Please, clarify.

Thank you for your comment, we have included in line 151-154 the following sentence: “Toddlers from 12 to 36 months of age, who were born with a healthy gestational age, weight, and height and who did not present pathological complications that interfered with their eating behaviors, in addition to being accompanied by the biological mother were included in the study.”

Who recruited the sample? How? Were the mothers compensated for their recruitment?

Thank you very much for your observation. We have included the following information in line 155-162: “A team of nutrition graduates from the University of Guadalajara collected data over a 6-month period at the NHCJIM outpatient clinic. They explained the study's objectives to the patients, obtained biological kinship and informed consent, and measured the mothers' anthropometric data. Then, they asked the mothers to complete the AEBQ-Esp and measured the anthropometric data for toddlers. Finally, they were asked to complete the CEBQ-T. In appreciation, the team provided information to improve toddler dietary habits and tools for healthy feeding practices.”

Why did the authors choose these measures over others?

Thank you very much for asking this question. Our study is based on the Behavioural Susceptibility Theory (BST), which proposes that appetite mediates the interaction between genetic susceptibility to obesity and exposure to an obesogenic environment, thus measured differences in appetite determine why some people overeat or undereat, and others do not, in response to environmental opportunity (https://doi.org/10.1098/rstb.2022.0223).

In order to test this theory, psychometric measures of appetite have been used in both children and infants, using the Child Eating Behaviour Questionnaire and the Baby Eating Behaviour Questionnaire respectively, developed for this purpose and systematically reviewed, including a meta-analysis, have quantified the associations between the appetitive traits these questionnaires measure and adiposity (https://doi.org/10.1111/obr.13169).

Our research group has been working with these measures for several years and we have validated them in our country. The corresponding author also worked on her doctoral thesis in the UK to develop this measure for use in adolescents and adults. Our aim is to be able to provide further evidence of the BST in a culturally and socially different population, as we find in the participants in the current study.

In the results section the alert "Error! Reference source not found" is present several times.

Thank you for pointing these out, we are sorry for these mistakes, and we have corrected them. They included hyperlinks, we have now taken then out.

My main concern, which applies both to the introduction and discussion section is that there is very little or none reference to the Developmental Psychopathology theoretical and clinical framework. In general, the manuscript lacks a developmentally oriented standpoint. Besides, the aspects associated with genetic issues are also scarcely addressed. I suggest adding information coming from the following references: https://www.cairn.info/revue-devenir-2004-3-page-173.htm ; https://doi.org/10.1002/imhj.21732 ; https://www.ncbi.nlm.nih.gov/pmc/articles/PMC4726243/

Thank you very much for your comments and input. These strengthens the theoretical background of the paper. We have added to the introduction section, as well as the discussion and the limitations as well:

In lines 88-90: “Also, the parent's affective and personality disorders could undermine the functioning of the caregiving relationship, resulting in unpredictable and inconsistent emotional communication with the child [17].”

In line 104-110: “Parents may also find it difficult to recognize their own styles of eating and disor-dered eating behaviors, that may have an impact on their child eating behaviors or their ability to diagnose their own child’s style of eating, mediating parent-infant interactions [17,23]. Other psychophysical aspects of eating behavior could also be playing a part in the influence of appetite on weight. For example, both pre and post-gestational differ-ences in maternal metabolism modify the mother’s eating behavior, which in turn could affect the infant’s appetite as well as taste and odor for different types of foods [24].”

In line 279-284: “The presence of affective and personality disorders in parents may disrupt the functioning of the caregiving relationship, leading to erratic and inconsistent emotional communication with their child [17], affecting developmental weight trajectories in children observed longitudinally [32]. However, no mothers with history of known personality or eating disorders were included in the study.”

In line 405-408: “…as well as a measure of a number of different complex variables involved in psycho-pathological risk through intergenerational transmission [17], or as risk factors for dis-ordered eating development through the life course [47].”

In line 417-421: “These results point towards a potential intergenerational transmission of appetite, which as explained by the BST, through the presence of certain appetitive traits with a strong genetic basis, they act as a mediator for the genetic influence on weight. However, these are not the only factors that are at play, given the small to moderate magnitude of the correlations presented.”

Comments on the Quality of English Language

The quality of English is good. I only saw minor typos and/or grammatical errors.

Thank you very much, we have checked the document thoroughly for mistakes and apologize for the presence of these issues.

Reviewer 2 Report

I have some observations on the information used, when referring to mother/toddler appetite, it is necessary to clarify some points:

Appetite is not the only risk factor associated with the mother that may predispose the toddler to develop overweight or obesity.  Throughout the document, the information is present as cause and effect and it is very risky.

Although bitter taste differs widely among individuals, at psychophysical and genetic levels, toddlers are more sensitive to the taste of some bitters, which sweet and salty taste can partially mask or block.

Similarly, pre- and post-gestational maternal metabolism modifies the mother's eating behavior and therefore, there are critical periods of development that may predispose infants to increase their appetite or taste flavors and odors.

This type of information is not referred to in the document. 

The age of the mothers who participated in the study is 18-40 years. I insist, the metabolic and neurological conditions associated with hunger and appetite are very different in the selected range.  I recommend making age ranges, just as they did with toddlers.

In addition to the above, their inclusion and exclusion criteria do not indicate whether they are primigravid mothers or the number of previous pregnancies, which is another factor to consider for both the mother and the offspring.

Early in life, children learn the rules of cuisine, how to eat, what to eat, when to eat, and what foods are supposed to taste like. In addition, this learning most often occurs in the context of the changing dynamic between mother and child and the food environment in which they live.

The results indicate that most of the infants did not develop obesity, however this is not disputed.

While the study is interesting, it requires refinement to make it more objective.

Therefore, it is very risky to say that there is an Intergenerational transmission of appetite associated with appetite and weight in toddlers with the methodological tools used

Author Response

I have some observations on the information used, when referring to mother/toddler appetite, it is necessary to clarify some points:

Thank you for your time in reviewing our study, we have tried to address the issues you have commented upon, and we have now included further information and evidence we hope has clarified and enriched the document.

Appetite is not the only risk factor associated with the mother that may predispose the toddler to develop overweight or obesity.  Throughout the document, the information is present as cause and effect and it is very risky.

Thank you very much for your observation. This is an issue we have now tried to address throughout the document and highlighted within. We hope we have been able to do so, and we have added these issues as important limitations to the manuscript as well.  

In line 79-83: “However, although there is sufficient evidence of the genetic link between appetite and weight variations, this is not likely the only pathway to express changes in weight, given that the correlations observed are small [4].  Many other factors could be contributing to weight trajectories in infants, toddlers, children and further playing a role in the inter-generational transmission of appetite and eating behaviors [17].”

Although bitter taste differs widely among individuals, at psychophysical and genetic levels, toddlers are more sensitive to the taste of some bitters, which sweet and salty taste can partially mask or block.

Thank you for your suggestion. We have added this information to the background and the discussion:

In line 85-89 “Genetic and environmental preferences for different foods have also been observed and could contribute to different weight trajectories in infants and children, with genetic effects playing a bigger role in the preference for sweeter tasting fruits, bitter tasting vegetables, and proteins in a large sample of 3-year-old British twins (n = 2686) [18].”

Similarly, pre- and post-gestational maternal metabolism modifies the mother's eating behavior and therefore, there are critical periods of development that may predispose infants to increase their appetite or taste flavors and odors. This type of information is not referred to in the document.

These are very pertinent comments, thank you. We have included some of these points in the paper in the introduction:

In line 104-110: “Parents may also find it difficult to recognize their own styles of eating and disordered eating behaviors, that may have an impact on their child eating behaviors or their ability to diagnose their own child’s style of eating, mediating parent-infant interactions [17,23]. Other psychophysical aspects of eating behavior could also be playing a part in the influence of appetite on weight. For example, both pre and post-gestational differences in maternal metabolism modify the mother’s eating behavior, which in turn could affect the infant’s appetite as well as taste and odor for different types of foods [24].”

The age of the mothers who participated in the study is 18-40 years. I insist, the metabolic and neurological conditions associated with hunger and appetite are very different in the selected range.  I recommend making age ranges, just as they did with toddlers.

Thank you for your comments. In order to address this issue, we have carried out further analysis which we included in table 5. We have included these in the results as well as the discussion and conclusion of the article:

In line 252-254: “These associations were not seen when maternal appetitive traits were separated by age categories.”

In line 264-269: “These associations were similarly observed by maternal age category in the 18–25-year-olds [“Food responsiveness” [r=0.25; p<0.001], “Emotional overeating” [r= 0.28; p<0.001], “Enjoyment of food” [r= 0.22 p<0.0001], “Satiety responsiveness” [r= 0.22; p<0.001] and “Food fussiness” [r= 0.25; p<0.001]], as well as for “Food responsiveness” [r=0.25; p<0.001], “Emotional overeating” [r=0.19; p<0.001] and “Enjoyment of food” [r=0.27; p>0.001] in the 31–40-year-olds.”

In line 289-295: “These same associations were not found when the mother’s age was categorized, except in those mothers who were the youngest (18-25 years of age) and also, the highest in number in our study and only some associations were found in the 31- to 40-year-old group, where only “Food responsiveness”, “Emotional overeating” and “Enjoyment of food” were positively correlated with their toddler’s counterpart appetitive traits. These differences could be due in part to metabolic and neurological conditions associated with appetite in different age groups of the mother [24].”

In line 381-385: “This is a novel study that measures appetite traits in mother-toddler dyads, which suggest there is the potential for an intergenerational transmission of appetite. These associations and the measurement of many other factors at play, not present in this study, could provide some evidence towards observed inherited variations in appetite [4] between mothers and toddlers. Thus, the study presents several limitations.”

In line 413-421: “This was true for all food approach appetitive traits such as “Food responsiveness”, “Emotional overeating”, “Enjoyment of food”; and two out of three food avoidance appetitive traits, “Satiety responsiveness”, and “Food fussiness”, except for “Slowness in eating”, when all maternal ages were taken into account, but not when they were cat-egorized by age. These results point towards a potential intergenerational transmission of appetite, which as explained by the BST, through the presence of certain appetitive traits with a strong genetic basis, they act as a mediator for the genetic influence on weight. However, these are not the only factors that are at play, given the small to moderate magnitude of the correlations presented.”

In addition to the above, their inclusion and exclusion criteria do not indicate whether they are primigravid mothers or the number of previous pregnancies, which is another factor to consider for both the mother and the offspring.

Thank you for your comment, we included this information in table 1 as birth order, however, we do not have the exact data of the number of pregnancies the mother had had.

Early in life, children learn the rules of cuisine, how to eat, what to eat, when to eat, and what foods are supposed to taste like. In addition, this learning most often occurs in the context of the changing dynamic between mother and child and the food environment in which they live.

Thank you for your comment. This information is mainly included in lines 92-110 that describe the role of parents and their relationship to appetite traits:

“Parental feeding practices are a key component of a child’s food environment. Evidence suggests that the use of nonresponsive feeding practices have a detrimental impact on appetite development, by encouraging children to eat for reasons other than hunger (i.e. using food as a reward for desired behavior or to soothe emotions) or may over-ride their satiety cues by pressuring them to eat [19,20]. Also, they may influence them in multiple ways, including via modelling of eating behaviours, however, the relationship between parent and child self-regulation of appetitive traits has received little attention [21]. Responsive feeding practices by mothers were better achieved when toddler’s had higher Satiety responsiveness and lower Food responsiveness [22]. When children have avid appetites, parents may also find it difficult to maintain a healthy food environment as their demands for certain foods to be present in the home can be stressful and occur as early as the introduction of family meals during early toddlerhood [19].

Parents may also find it difficult to recognize their own styles of eating and disordered eating behaviors, that may have an impact on their child eating behaviors or their ability to diagnose their own child’s style of eating, mediating parent-infant interactions [17,23]. Other psychophysical aspects of eating behavior could also be playing a part in the influence of appetite on weight. For example, both pre and post-gestational differences in maternal metabolism modify the mother’s eating behavior, which in turn could affect the infant’s appetite as well as taste and odor for different types of foods [24].”

The results indicate that most of the infants did not develop obesity, however this is not disputed.

Thank you very much for your observation. Mexican health and nutrition surveys ENSANUT 2018, show that the probability to overweight or obesity in Mexican population increase at 5 years, this age corresponds to elementary school where they have more access to ultraprocessed food, while in the toddler stage they depend mainly on foods that are offered by the caregivers. For this reason, we included the following reference in line 316-318: “This may indicate that at the toddler stage, external obesogenic factors such as the food environment, have not yet had a significant influence on the toddler’s eating behaviours and therefore it is not reflected in the toddler’s weight status [16].”

As this is a cross-sectional study, we do not know the weight trajectories of the toddlers, and we cannot see changes in weight, this would require birth weight measurements which we did not have.

While the study is interesting, it requires refinement to make it more objective.

Thank you for your suggestions. We have tried to add to the study to make it more objective as suggested. Several comments are highlighted throughout the document in different sections. We hope you find these improve the final paper.

Therefore, it is very risky to say that there is an Intergenerational transmission of appetite associated with appetite and weight in toddlers with the methodological tools used.

We have also made a suggested change to the title of the paper. We hope this is more suitable.

We have changed the title of the paper from “Intergenerational transmission of appetite: Are maternal appetitive traits associated with appetite and weight in toddlers?” to:

“Association between maternal and toddler appetitive traits in a Mexican population.”

Reviewer 3 Report

This study assessed appetite traits in mother-toddler dyads, and based on the obtained results, their primary conclusion centered on the concept of intergenerational transmission of appetite."

Major concern: This is a very interesting article that provides guidance for better parenting. I really like this article. However, I have some reservations about the article's conclusion, although I would like to support the author's viewpoint. I believe that the author's evidence can only demonstrate mutual influence between mothers and infants during the breastfeeding period. If the author wants to draw the conclusion of 'intergenerational transmission of appetite,' more crucial data is needed: the mother's appetite traits before pregnancy. I would support the author's conclusion only if the mother's appetite traits before pregnancy are similar to those during the breastfeeding period."

Minor:

1. Could you please provide explanations for the abbreviations 'AEBQ-Esp,' 'CEBQ-T-Mex,' and 'BMIz' when first introducing them?

2. In the introductory paragraph, the article you referenced is not the first one to propose the concept of BST. You should cite the article that initially proposed this concept."

3. Line 80. “<.001” should be “p < .001”.

4. Line 138. “mother’s” should be “mothers’”.

5. Line141-148. It would be more consistent to place the item in quotation marks and not use the semicolon.

6. Line 147. “strongly disagree” should be “strongly agree”.

7. Line 169. “12-24 and 24-36 years”, I think should be “12-24 and 24-36 months”.

8. Line 177-178, 186-188 and 199. What does “Error! Reference source not found.” mean?

9. Line 190. “that” should be “than”.

This manuscript appears to be clear and mostly free of grammatical errors. Only some small parts lack clarity in reference and are not concise enough: Line 164. The phrase "On the other hand" could be used to introduce a contrasting point. However, in this context, it seems more like an extension of the previous point. Consider rephrasing for clarity, such as "In addition," or "Furthermore." Line 165. it's unclear what "these children" refers to.

Author Response

This study assessed appetite traits in mother-toddler dyads, and based on the obtained results, their primary conclusion centered on the concept of intergenerational transmission of appetite."

Thank you for the time you have taken to review this article, we appreciate all of your comments and hope we have been able to address them.

Major concern: This is a very interesting article that provides guidance for better parenting. I really like this article. However, I have some reservations about the article's conclusion, although I would like to support the author's viewpoint. I believe that the author's evidence can only demonstrate mutual influence between mothers and infants during the breastfeeding period. If the author wants to draw the conclusion of 'intergenerational transmission of appetite,' more crucial data is needed: the mother's appetite traits before pregnancy. I would support the author's conclusion only if the mother's appetite traits before pregnancy are similar to those during the breastfeeding period."

Thank you very much for pointing out your concerns. We hope the highlighted changes we have made to the text can help address your concerns. Included in these, are changes in the title of the study from “Intergenerational transmission of appetite: Are maternal appetitive traits associated with appetite and weight in toddlers?” to:

“Association between maternal and toddler appetitive traits in a Mexican population.”.

Minor:

  1. Could you please provide explanations for the abbreviations 'AEBQ-Esp,' 'CEBQ-T-Mex,' and 'BMIz' when first introducing them?

Thank you for your suggestions. We apologize for not clarifying these at the beginning of the document. We have now included them in the document as well as the abstract.

  1. In the introductory paragraph, the article you referenced is not the first one to propose the concept of BST. You should cite the article that initially proposed this concept."

Thank you for your comment. We have included the initial paper by Carnell https://pubmed.ncbi.nlm.nih.gov/18614720/ , as well as a following paper by Llewellyn and Fildes https://www.ncbi.nlm.nih.gov/pmc/articles/PMC5359365/ . We hope these provide better evidence for the BST.

  1. Line 80. “<.001” should be “p < .001”.

Thank you, we have made the corresponding changes.

  1. Line 138. “mother’s” should be “mothers’”.

We have also made this change, thank you for pointing it out.

  1. Line141-148. It would be more consistent to place the item in quotation marks and not use the semicolon.

Thank you for your comment. We wanted to show how many items made up each subscale. We have changed the semicolon for a colon, we hope this makes it more consistent.

  1. Line 147. “strongly disagree” should be “strongly agree”.

We have made the change, thank you for noticing.

  1. Line 169. “12-24 and 24-36 years”, I think should be “12-24 and 24-36 months”.

We have made the change, thank you for noticing.

  1. Line 177-178, 186-188 and 199. What does “Error! Reference source not found.” mean?

We apologize for the mistakes, the text had hyperlinks. We have corrected these.

  1. Line 190. “that” should be “than”.

Thank you, we have made the change.

Comments on the Quality of English Language

This manuscript appears to be clear and mostly free of grammatical errors. Only some small parts lack clarity in reference and are not concise enough: Line 164. The phrase "On the other hand" could be used to introduce a contrasting point. However, in this context, it seems more like an extension of the previous point. Consider rephrasing for clarity, such as "In addition," or "Furthermore."

Thank you for your comments. The text has been altered substantially; we hope we have addressed these issues as well.

Line 165. it's unclear what "these children" refers to.

Thank you for your observation. We have cleared up this mistake.

Round 2

Reviewer 2 Report

  • I have no comment.  The authors addressed all the comments made in the original document. 
  •